# A Promising Future for Stem-Cell-Based Therapies in Muscular Dystrophies—In Vitro and In Vivo Treatments to Boost Cellular Engraftment

**DOI:** 10.3390/ijms20215433

**Published:** 2019-10-31

**Authors:** Daniela Gois Beghini, Samuel Iwao Horita, Liana Monteiro da Fonseca Cardoso, Luiz Anastacio Alves, Kanneboyina Nagaraju, Andrea Henriques-Pons

**Affiliations:** 1Laboratório de Inovações em Terapias, Ensino e Bioprodutos, Instituto Oswaldo Cruz, Fiocruz, Rio de Janeiro (RJ) 21040-900, Brazil; beghini@ioc.fiocruz.br (D.G.B.); samuel.horita@ioc.fiocruz.br (S.I.H.); 2Laboratório de Comunicação Celular, Instituto Oswaldo Cruz, Fiocruz, Rio de Janeiro (RJ) 21040-900, Brazil; lianamfc@gmail.com (L.M.d.F.C.); alveslaa@gmail.com (L.A.A.); 3Department of Pharmaceutical Sciences, School of Pharmacy and Pharmaceutical Sciences, Binghamton University, New York, NY 13902, USA; knagaraju64@gmail.com

**Keywords:** muscular dystrophy, stem cells therapy, muscle regeneration, matricryptins and matrikines, inflammation

## Abstract

Muscular dystrophies (MD) are a group of genetic diseases that lead to skeletal muscle wasting and may affect many organs (multisystem). Unfortunately, no curative therapies are available at present for MD patients, and current treatments mainly address the symptoms. Thus, stem-cell-based therapies may present hope for improvement of life quality and expectancy. Different stem cell types lead to skeletal muscle regeneration and they have potential to be used for cellular therapies, although with several limitations. In this review, we propose a combination of genetic, biochemical, and cell culture treatments to correct pathogenic genetic alterations and to increase proliferation, dispersion, fusion, and differentiation into new or hybrid myotubes. These boosted stem cells can also be injected into pretreate recipient muscles to improve engraftment. We believe that this combination of treatments targeting the limitations of stem-cell-based therapies may result in safer and more efficient therapies for MD patients. Matricryptins have also discussed.

## 1. Muscular Dystrophies

Muscular dystrophies (MDs) are genetic disorders caused by mutations in several genes that lead to the lack of or dysfunctional production of proteins that are essential for myofiber integrity and contraction. MDs are a group of diseases that cause, but are not restricted to, progressive muscle destruction and weakness, with nine most common forms: myotonic, Duchenne, Becker, limb-girdle, facioscapulohumeral, congenital, oculopharyngeal, distal, and Emery–Dreifuss. Each disorder varies significantly in severity, pattern of inheritance, age of onset, affected gender, targeted muscles and other organs, levels of muscle damage, etc. (summary in Appendix A). The approximate combined prevalence of all MDs ranges between 19.8 and 25.1 per 100,000 people/year, a very high incidence of debilitating diseases that affect social integration and life expectancy [1]. For a conclusive diagnosis, it is essential to observe medical and family history, including the distribution of weakness in different muscles and the age of onset, gene sequencing, laboratory investigations, electromyography, and muscle biopsy. Prognosis is also highly variable across MD patients, varying from mild to severe disability and early death, usually with the progressive development of symptoms. Comprehension of MD pathophysiology has increased considerably in the last decade, but there remains no cure. Current treatments, in general, aim to manage and slow the progression of symptoms, which reinforces the need for new therapies to increase patients’ life expectancy and quality of life.

In this review, we have summarized the most common forms of MD and the therapies available. However, most patients receive only palliative therapeutic strategies that aim to alleviate the symptoms, with no effective treatments. Although stem-cell-based therapies offer hope for these patients, they have many limitations. Stem cells usually have a limited capacity to engraft muscles due, for example, to reduced cellular viability, dispersion, proliferation, and differentiation to myotubes. Many of these limitations have been individually addressed in the literature, and there are biochemical, genetic, and in vitro culture approaches that can improve cellular engraftment. Here, we have summarized these studies and propose combined treatments for stem cells and their recipient muscles, aiming to prolong the survival of grafted cells and promote sustained, healthy myotube formation. Although not yet tested in muscle stem cells, matricryptins are nontoxic, bioactive peptides that induce tumor cell survival, migration, proliferation, and differentiation, all desired biological responses for muscle stem cells. These molecules are possible boosts that could be used in combination with other stimuli for stimulating stem cells in muscle therapy. The broad spectrum of pathogenic genetic defects and clinical symptoms suggests that different combinations of boosts may be required for different MDs.

### 1.1. Myotonic Muscular Dystrophy (MMD)

Myotonic muscular dystrophy (MMD) is a dominant, autosomal disease with two similar but distinct forms, type 1 (MMD1) and 2 (MMD2). MMD1 is also named Steinert’s Disease, in honor of the scientist who first described the disease in 1909. MMD2 was first described more recently thanks to genetic testing becoming available for clinical practice. MMD1 is the most common form of the disease and is found in adults, although a late onset is not a rule. The symptoms include muscle weakness and wasting, cardiac conduction defects, myotonia, diabetes and insulin resistance, cataracts, and many others, leading to multisystem involvement. MMD1 can be divided into four different forms according to clinical phenotype, illustrating the broad range of symptoms and general characteristics [2]. In adult-onset MMD, diagnostic efforts are usually initiated because of muscle weakness, myotonia, or cataracts—the three main symptoms. In this case, a family history of type 1 MDD combined with minor symptoms is a common starting point in diagnostic examination. The progression of the disease is slow and advances to deepened skeletal muscle weakness, including the face, neck, and distal limb muscles; eventual immobility; respiratory insufficiency; dysarthria; and dysphagia. The latter is one of the leading causes of severe disability and death in the late stages of adult-onset MMD1. The cardiac muscle shows conduction defects and tachyarrhythmia, most likely due to fibrosis. Usually, the detection of cataracts in older patients does not initiate further diagnostic considerations for MMD1, but should particularly alert clinicians when detected in patients under the age of 50 and associated with specific structural characteristics [3]. The central nervous system can also be affected, leading to progressive cognitive impairment and late apathy [4], as well as gastrointestinal disorders such as constipation, incontinence, and diarrhea [5]. Endocrine dysfunction is also found, leading mostly to insulin resistance and susceptibility to diabetes, hypothyroidism, male hypogonadism, and adrenal insufficiency [6]. The second form of MMD1 is Congenital MMD; this is the most severe form of MMD, and is usually detected prenatally because of reduced fetal movements and different deformities. At birth, babies have severe hypotonia in their limb, trunk, respiratory, facial, and bulbar muscles, leading to respiratory dysfunction and feeding difficulties [7]. Mental retardation can also be observed. In childhood-onset MMD, there is no myotonia or muscle weakness, which imposes challenges for MD diagnosis and correct assistance. Instead, children have delayed learning at school and show signs of mental retardation. These patients also develop muscle weakness and wasting at an older age, causing physical disabilities comparable with severe adult-onset type 1 disease [8]. Finally, in late-onset oligosymptomatic MMD, the genetic family history is significant for clinicians considering a MMD diagnosis, as mild symptoms can be observed in earlier generations, like cataracts and discrete muscle weakness. However, later generations can have severe disease characteristics, including muscle wasting and atrophy, cataracts, and others [2].

The broad spectrum of tissues and organs affected, not restricted solely to muscles, and the disease severity in MMD1 is determined by the underlying molecular pathogenesis which is mediated by deleterious nuclear RNA repeats. Therefore, abnormalities in many pathways of RNA metabolism, including alternative splicing, can be detected in MMD1. In healthy cells, pre-mRNA splicing is mediated by nuclear protein factors that regulate the processing of mature RNA and the translation of functional proteins. In healthy individuals, there are low numbers of a CTG repeat in the 3′-UTR untranslated region of the dystrophia myotonica protein kinase (*DMPK*) gene (Figure 1A). However, in MMD1, there is an unstable expansion of this repeat [9], leading to *DMPK* transcripts that have expanded CUG repeats (CUG_n_DMPK), which are retained and distributed in the nucleus as discrete, visible foci. These structures sequester double-stranded CUG-binding proteins, including the best-characterized player in this mechanism, the muscleblind-like (MBNL) family of splicing factors (Figure 1B). These proteins interact physically with expanded CUG repeats of the *DMPK* RNA and drastically reduce its availability for unprocessed RNA strains. One of the main pre mRNA targets of the CUG_n_DMPK + MBNL is the chloride channel RNA construct (*CLCN1*), which leads to aberrantly spliced *CLCN1* transcripts with additional exons and premature termination codons, cytoplasmic degradation through the nonsense-mediated decay pathway, truncated nonfunctional CLCN1 protein, and/or dysfunctional channel activity [10] (Figure 1B). All these alterations lead to the increased muscle excitability observed in MMD patients.

It is known that the number of repeated CUG sequences is directly related to disease severity, with 38 to 50 repeat sizes being considered premutations and generally not leading to apparent symptoms. On the other hand, mild phenotypes are associated with sequences with 51 to 149 repeats, and early, more severe onset phenotypes have more extended repeat sequences, with at least 1000 repeats [11]. The distribution of affected tissues and organs is at least partially explained by somatic mosaicism due to repeat size instability during mitosis. Peripheral blood leukocytes, for example, have smaller expansion sizes when compared with other cell types and cells from other tissues [11].

Although there are many overlapping phenotypic features in both types of MMD, key characteristics that distinguish MMD2 from MMD1 include more proximal muscle weakness and general milder cardiac and multisystem involvement in MMD2 [12]. In contrast to MMD1, MMD2 onset occurs in adulthood, with no reports of congenital development. In MMD2, there is a (CCTG)_n_ expansion in intron 1 of the cellular nucleic acid binding protein/zinc finger protein 9 (CNBP/ZNF9) [13] and sequestration of MBNL in the hairpin of repeats [14], but the pathophysiology of MMD2 is not well understood [15].

There is no cure for MMD, but studies of new therapeutic strategies focus mostly on reducing RNA repeat sequences and preventing interactions of the RNA hairpins of repeated sequences with MBNL. Antisense oligonucleotides for the CUG repeats in the *DMPK* transcripts are especially promising, with the potential to discriminate between normal and mutated transcripts. Moreover, many alternative therapeutic strategies using small chemicals that upregulate MBNL and modulate protein kinases, among other strategies, are under development with good results (reviewed by López-Morató et al. [10]).

### 1.2. Oculopharyngeal Muscular Dystrophy (OPMD)

OPMD is a late-onset neuromuscular disorder, with the initial symptoms of lowering of the eyelids (ptosis) and swallowing difficulties (dysphagia). With the progression of the disease, other skeletal muscles can be affected, including the proximal muscles of the lower limbs. In general, the symptoms of OPMD are similar to myasthenia gravis, for example, and clinicians must pay special attention to some pathological hallmarks, like the formation of insoluble inclusions in the nuclei of muscle cells [16]. OPMD is a monogenic disorder, with a mutation in the gene encoding for poly-adenylate (poly(A)) binding protein nuclear 1 (PABPN1) [17], leading to a short GCG expansion in its polyalanine tract. This protein is a multifunctional regulator of RNA metabolism [18,19,20] and, despite its ubiquitous expression, mutated PABPN1 leads to symptoms manifested predominantly in skeletal muscles, where the levels of protein expression are lower [21]. It is not yet clear why the disease initiates from midlife onward, nor is the precise nature of the correlation between mutated PABPN1 levels and the poly(A) tail length, or how the protein regulates changes in RNA metabolism. Current treatments for OPMD are limited to surgical corrections for ptosis and dysphagia targeting the cricopharyngeal muscle, although some molecular therapies are under investigation [22,23].

### 1.3. Emery–Dreifuss Muscular Dystrophy (EDMD)

EDMD was characterized by its clinical features and disease course in 1966 [24]. It affects mainly the brachial and fibular muscle groups, induces multiarticular contractures and spine rigidity, and induced cardiomyopathy with conduction disturbances later on [25,26,27]. The first symptoms appear in the first decade of life, manifesting as ankle and elbow contractures and spine rigidity. In the second and third decade of life, muscle atrophy and weakness are more visible, usually with a slow progression [28]. EDMD is divided into at least four types (1 to 4), and EDMD1 is associated with mutations in the emerin gene (*EMD*) located on the X chromosome, a protein that spans the inner nuclear membrane and regulates several nuclear functions (Figure 2). Emerin binds to lamins A and C to form the nuclear lamina, a proteinaceous network that regulates the architecture and function of nuclear DNA. Emerin regulates gene expression by binding to and regulating the activity of many transcription factors and downstream signaling pathways, including the regulation of histone deacetylase 3 (HDAC3). This enzyme alters the chromatin structure and regulates transcription factor activity. The *EMD* gene contains six exons, and the first mutation associated with EDMD identified was c.3G > A, affecting codon start in exon 1. Since then, the nonsense mutation c.130C > T in exon 2 and c.653insTGGGC in exon 6 have also identified, all influencing an open reading frame that creates a premature stop codon at the 238 position [29]. In most EDMD1 cases, small deletions or splice-site mutations lead to this codon change, while the remaining patients have nonsense/missense mutations or large deletions [30].

In EDMD2, symptom severity ranges extensively, with muscle atrophy, joint contractures, and loss of ambulance occurring early in some patients, in contrast to patients with mild and late onset, which are associated with slow progression [31]. Respiratory muscle weakness and chest deformities usually lead to respiratory failure, while left ventricle systolic dysfunction and cardiac conduction disturbances lead to dilated cardiomyopathy and sudden death. The autosomic *LNMA* gene has been identified by mapping to be responsible for EDMD2; it is located in the long arm of the chromosome 1 at position 22(1q22) and encodes the lamins, mainly lamin A and C (Figure 2). In most patients, genetic alterations involve heterozygous missense mutations and generate lamins A [32] and C with a dominant-negative toxic effect, although some deletions or duplications may lead to not functional proteins [33].

The identification of other genes related to the organization of nuclear structures interfacing the chromatin has led to the description of different types of EDMD. For example, mutations in the *SYNE1* and *SYNE2* genes (the synaptic nuclear envelope protein 1(or 2)), encoding for nesprin-1 and nesprin-2, are linked to EDMD4 and EDMD5, respectively [34]. Another example is the *LAP* gene, which encodes polypeptides connected with lamins (lamin-associated protein 2 alpha, LAP2alpha) [35] and mutations in the *FHL-1* (four and a half LIM domains protein 1) gene located on the X chromosome [36]. Unfortunately, EDMD treatment also relies mostly on targeting symptoms, including surgical procedures.

### 1.4. Limb-Girdle Muscular Dystrophy (LGMD)

LGMDs are an extremely heterogeneous group of autosomal dominant or recessive disorders that, in general terms, lead to weakness and wasting of muscles in the legs and arms. LGMDs are usually not syndromic and are mostly limited to skeletal muscles. However, some patients have cardiac problems [37], and a few patients exhibit evolutionary delays and mental disabilities [38]. LGMD patients have highly variable phenotypes and clinical courses, and can be affected at any age [39]; however, one of the few unifying characteristics is that the symptoms are more severe when the disease manifests early in life [40]. The phenotypic heterogeneity is a reflection of the multiple genes that can be affected in LGMD—genes involved in muscle fiber integrity, repair, and function, producing abnormal proteins that are located in the sarcomere, sarcolemma, extracellular matrix, nucleus, and other places [41].

As reviewed by Taghizadeh [42], more than 30 different subtypes of LGMD have been described so far. The most common subtypes are LGMD1A (myotilinopathy), LGMD1B (laminopathy), LGMD1C (caveolinopathy), LGMD1E and LGMD2R (desminopathies), LGMD2A (calpainopathy), LGMD2B (dysferlinopathy), LGMD2C, 2D, 2E, and 2F (the sarcoglycanopathies), LGMD2G (telethoninopathy), LGMD2I, 2K, 2M, 2N, 2O, 2P, 2T, and 2U (the dystroglycanopathies), LGMD2J (titinopathy), and LGMD2L. Figure 2 summarizes some of the LGMD-associated proteins and illustrates the wide range of muscular functions affected by mutations in the disease.

### 1.5. Facioscapulohumeral Muscular Dystrophy (FSHD)

FSHD is one of the most frequent MDs and affects mainly the muscles of the face, shoulder blades, and upper arms, but muscle weakness is usually progressively observed in other muscles. The first signs start before the age of 20, with weakness and discrete atrophy of the muscles around the mouth and eyes, shoulders, lower legs, and upper arms, and abdominal and hip muscles can be affected later. FSHD progresses very slowly, rarely affecting the cardiac or respiratory system, and most patients have an average life span [43,44].

Most FSHD type 1 patients have a genetic mutation leading to the pathogenic loss of D4Z4 microsatellite repeats [45], and each D4Z4 unit on chromosome 4q35 contains a copy of the *DUX4* (double homeobox 4) retrogene, a germline transcription factor [9]. In healthy individuals, 11 to 100 D4Z4 repeats (each repeat with 3.3 kb) are observed, with normal chromatin methylation in most cases, and no transcription of *DUX4* in somatic tissues. However, FSHD patients have a contraction in the number of repeats, varying from 1 to 10 repeats, with the reduction in the number of repeats associated with disease severity. The presence of at least one repeat containing a copy of the *DUX4* gene is required for FSHD development [46], leading to hypomethylation in most cases and *DUX4* transcription (decreased repression of the D4Z4 repeat). However, it seems that the transcribed full-length *DUX4* mRNA is unstable due to the lack of a polyadenylation signal sequence, and much remains to be understood about the pathophysiology underlying the FSHD phenotype [47]. Moreover, mutations in the genes *SMCHD1* or *DNMT3B* are associated with FSHD type 2, also leading to aberrant expression of *DUX4* in skeletal muscles and similar symptoms to FSHD type 1.

### 1.6. Congenital Muscular Dystrophy (CMD) and Distal Muscular Dystrophy

CMDs are a highly heterogeneous group of very early-onset severe muscle diseases with symptoms that include decreased mobility, delay or arrest of gross motor development, and joint or spinal deformities that impose significant difficulties on patients’ development. With the progression of the disease, cardiac and respiratory complications may occur and, in some subtypes, the central nervous system and connective tissue can be affected [48,49,50]. A final diagnosis of CMD requires an integrated clinical and pathological analysis [38] because of a relatively overlapping spectrum of symptoms spanning characteristics of congenital myopathies, congenital myasthenic syndromes, LGMD, and others [48]. There are more than 13 genes associated with CMD, and the primary disease subtypes are caused by laminin alpha-2 (merosin) deficiency (MDC type 1A, MDC1A) or partial merosin deficiency (MDC1B), fukutin-related proteinopathy (MDC1C), or acetylglucosaminyltransferase-like protein (LARGE)-related CMD (MCD1D) [51]. Other affected proteins include collagen VI, integrin α7, selenoprotein N, fukutin, O-mannosyltransferase 1 (POMT1) (Figure 2), and many others, generating a complex disease from the perspective of diagnosis and treatment.

Distal myopathies are also clinically, pathologically, and genetically heterogeneous, with highly variable phenotypes. In contrast to CMD, the age of onset ranges from childhood to late adulthood, and muscle weakness usually initially affects very distal muscles, like the finger and toe extensor muscles. As the disease progresses, proximal muscles may become impacted, but the distal weakness remains the most severe. The majority of distal myopathies are genetically determined, although acquired myopathies can occasionally manifest with distal weakness (e.g., sporadic inclusion body myositis, sarcoid myopathy, or focal myositis [52]). Mutations in many different genes encoding for proteins such as caveolin-3, dysferlin, α-actin-1, myotilin, desmin, and many others lead to distal muscular dystrophy (reviewed in Reference [53]). In both CMD and distal muscular dystrophy, palliative approaches address the alleviation of symptoms and there is no cure.

### 1.7. Duchenne Muscular Dystrophy (DMD) and Becker Muscular Dystrophy (BMD)

DMD is an X-linked disorder and the most common form of MD, affecting one in 3600 to 6000 male births. It is characterized by progressive muscle wasting, with disease onset around three years of age and patients becoming wheelchair-bound around their teens. Cardiorespiratory complications become progressively more severe, and death usually occurs in the second decade of life. Proximal muscle groups are the most affected muscles, leading to Gower’s sign, when the child climbs up on their own body to reach an upright posture. Calf enlargement is also observed as a sign of muscle inflammation and fat deposition [54,55]. Increased blood CK and hepatic transaminase levels combined with specific clinical signs and biopsy or DNA testing are used to make a diagnosis of DMD [55]. DMD is caused by mutations, mainly deletions, altering the open reading frames of the dystrophin gene, which has 79 exons, resulting in the absence of dystrophin [56]. This intracellular protein connects the cytoskeleton to the DGC, providing the link between the intracellular and extracellular environments (Figure 2). The lack of dystrophin leads to sarcolemma disruption, mainly due to mechanical damage, and local inflammation. With continuous cycles of damage and repair, the myogenic cells become exhausted, and muscle regeneration is compromised. In later stages of the disease, the accumulation of muscle fibrosis further deteriorates muscle function [57]. There is no cure for DMD, and the management of patients is multidisciplinary, involving general and respiratory physiotherapy, genetic counseling, and corticosteroid treatment. This pharmacological agent is currently used to improve muscle strength, cardiorespiratory function, and life quality and expectancy. Despite its efficacy, corticosteroid-based therapy has many side effects that are detrimental to patients [58]. Several new therapy proposals are emerging, aiming to improve the life span of people with DMD [59]. These new strategies aim to genetically restore dystrophin expression in vivo [60] to stabilize the sarcolemma and enhance the expression of compensatory proteins, such as utrophin, decrease inflammation, and improve muscle regeneration [61,62].

BMD is a mild phenotype of dystrophinopathy, with about one-third of the incidence of DMD. The underlying pathological difference between the two diseases is that the mutations in the dystrophin gene of BMD do not abort the protein’s expression but partially affect its function, leading to a milder clinical phenotype. BMD patients usually show symptoms later, around 12 years of age, with a higher incidence of cardiac dysfunction and death typically around the fourth decade of life [63,64].

The broad spectrum of pathogenic mutations and clinical symptoms associated with no curative therapies reinforce the necessity of the search for new strategies to treat MD patients. Stem-cell-based therapies are a promising alternative for most of these patients, and many efforts have been made to overcome the numerous limitations of these procedures.

## 2. General Concepts for Stem-Cell-Based Thearapies

Cellular therapy has emerged in the field of regenerative medicine with the general goal of rescuing the function of tissues or organs affected by different diseases [65]. Stem cells are preferable for therapeutic applications because they can self-regenerate, undergo unlimited proliferation, differentiate into several different cell types, are located in the body in pools of a few specific cells, and directly enable tissue regeneration in vivo [66]. Cellular therapies for MD are based on the delivery of healthy precursor cells into damaged muscles to contribute to regeneration and muscle function improvement.

Stem-cell-based therapies are becoming more extensively studied in the field of DMD, and several approaches using different cellular subpopulations of stem cells and treatments have been published to treat these patients, and the general concepts also apply to other MDs. Two fundamental routes exist for stem-cell-based therapies, namely the use of healthy donor cells expressing the correct gene, or the expansion of cells obtained from the patients (autografts), expressing the corrected gene or not. In both cases, there are advantages and disadvantages, and future challenges lie in identifying the positive biological responses of each approach and neutralizing the negative aspects of the numerous published proposals, aiming to unify the current knowledge into practical therapeutic management strategies. The most attractive alternative would be to isolate cells from the patient’s most accessible anatomical site, genetically correct the mutated gene in vitro, expand and treat the cells to improve engraftment, and re-transplant them [67]. The ideal cellular subpopulation to be used should have myogenic potential, be able to fuse with the recipient’s myofibers, and fill out the niche of host stem cells for continuous production of cellular progeny and sustained regeneration. Another required feature is the delivery of the cells into the correct muscles, either through systemic administration using cells that retain the ability to home to the required muscle regeneration sites, or by local muscle injection [68]. In fact, after systemic administration, transplanted cells mostly accumulate in the liver, spleen, kidneys, and lungs, and very few reach the damaged muscles. On the other hand, after local injection, transplanted cells usually fail to populate the muscles, mostly due to reduced survival rates and limited ability to migrate and spread within the injected muscles [69]. All these challenges can be overcome if we consider the advances achieved in the field of stem cell biology, and it is conceivable to imagine that the limitations of cellular transplantation therapies could be appropriately addressed in future trials.

Patients suffering from diseases that affect more restricted muscle groups are the most likely to benefit from stem cell therapies. To this end, as indicated in Figure 3, we propose two different stages of treatments, first at the cellular level to correct genetic mutations, prolong cell survival, increase cellular mobility along the endomysium, favor the proliferation of corrected healthy stem cells, and increase muscle cell fusion for regeneration; second, genetically corrected/treated isogenic stem cells could be co-injected with other cell types in chemically pretreated target muscles (Figure 3) to improve muscle cell engraftment and provide sustainable normal muscle function.

It is important to highlight that the immune system imposes a significant barrier to any therapy that aims to induce the production of proteins lacking in patients with genetic diseases, such as DMD. Additional studies must be conducted to delineate specific strategies able to silence the recognition or effector response of T cells against introduced proteins, with limited suppression of the adaptive immune system. However, in many MD subtypes, the mutated proteins are dysfunctional, retained in specific cellular compartments such as the nucleus, degraded, or produced at aberrant levels. In these cases, central and peripheral immunological tolerance mechanisms avoid self-damage after the genetic correction for normal protein expression with no immunological damage [70], in contrast to patients whose normal muscle function relies on the introduction of unexpressed proteins.

Pioneering studies have employed myoblast transplants for the development of new or hybrid muscle fibers containing dystrophin, in the particular case of DMD [71]. Myoblast transplantation was proposed in the early 1970s as a potential treatment for the disease, based on results obtained in *mdx* mice. In this murine model, the expression of dystrophin was restored by intramuscular injection of normal myoblasts; these results were followed by clinical trials in DMD patients who received intramuscular injections of allogeneic myoblasts [72]. All subsequent clinical trials using myoblast transplantation were mostly unsuccessful in terms of clinical benefit for the patients. It is likely that this inefficiency was due to the lack of adequate treatment with immunosuppressants, and also due to transplanted cell death and inefficient dispersion along the muscles, limiting the dissemination of the injected cells [73].

In DMD, where skeletal muscle weakness is disseminated, and other MDs involving various groups of skeletal muscles in the body, local delivery through intramuscular injection may not be a practical option. Therefore, systemic routes for cell delivery would be the ideal approach, such as intravenous injection of cells. However, this may lead to cell entrapment during the cells’ passage through organs (such as the lungs) [74], and may induce unwanted side-effects. However, when systemic routes are considered for transplantation, the cells should be able to bind to the endothelium, cross the barrier towards the interstitial muscle space, and eventually engraft into the damaged muscle fibers [75]. To this end, transplanted cells must express the required repertoire of molecules to be recognized by the cognate receptors presented by activated endothelial cells for transmigration. Therefore, the phenotype of transplanted cells is a crucial prerequisite that must be evaluated by clinicians before trials, considering the capacity of circulating cells to reach the target tissues.

## 3. Progenitor Cells Found in Skeletal Muscles and Stem Cell Thearapies

Regarding the subpopulations of stem cells found in skeletal muscles, the satellite cells are the first choice for cell therapy in MD. These cells are considered the primary cell type responsible for muscles growth and maintenance. However, the pioneering work published by Grounds [76] demonstrated that the number of resident satellite cells in healthy adult muscles is much smaller than the number of committed myogenic precursors populating the muscle tissue soon after injury. Afterward, several cellular markers were shown to identify and characterize muscular and non-muscular multilineage stem cells able to actively participate in myogenesis [77]. In this way, other cell types besides the satellite cells with myogenic potential, such as muscle-derived stem cells (MDSC) [78], mesoangioblasts [67,79], muscle-derived CD133^+^ progenitors [80], mesenchymal stem cells (MSC) [81], and PW1 interstitial cells [82], for example, have been described.

MDSCs have been identified to have the ability to self-renew and differentiate into mesodermal cells. Tamaki et al. [78] showed that single skeletal-muscle-derived CD34^−^/CD45^−^ (skeletal-muscle-derived double negative (Sk-DN)) cells exhibit clonal multipotency and can give rise to myogenic, vasculogenic, and neural cell lineages after in vivo single-cell-derived, single-sphere implantation, and in vitro clonal single-cell culture [78]. The correlation between other reported MDSCs is of interest, but it is difficult to directly compare them due to their high variability after muscle isolation and purification methods [78]. The main question regarding these cells is whether they have better ability to regenerate skeletal muscles than satellite cells. Among the MDSCs, Rouger et al. [83] isolated poorly adherent canine stem cells and investigated the efficacy of their systemic delivery in a clinically relevant DMD animal model to assess their potential therapeutic application. The cells were named MuStem cells (muscle stem cells) and were isolated from healthy dog muscles. When injected into immunosuppressed dystrophic dogs, they contributed to myofiber regeneration, satellite cell replenishment, and dystrophin expression. Importantly, their systemic delivery by intra-arterial injection resulted in long-term dystrophin expression with increased regeneration activity and persistent stabilization of the dog’s clinical status [83]. Although additional studies are necessary for a better understanding of these cells, they represent an alternative source of cellular therapies in humans, mainly because they easily proliferate in vitro.

Mesoangioblasts (MABs) are vessel-associated [84] activated cells derived from pericyte cells when isolated from postnatal tissues. Some cellular therapy studies have transferred MABs by the intra-arterial route into animal models of dystrophinopathies. These treatments resulted in significant recovery of dystrophin expression and normal muscle morphology and function, confirmed by contraction force measurement of single fibers. The outcome was usually a significant clinical amelioration and preservation of active motility [85]. MAB-like cells derived from induced pluripotent stem cells (iPSC) were expanded and genetically corrected in vitro with a lentiviral vector carrying the gene encoding the human α-sarcoglycan protein. When these cells were transplanted, they generated muscle fibers that expressed the target protein. Finally, transplantation of mouse iPSC-derived MAB-like cells resulted in functional amelioration of a dystrophic phenotype and restoration of the depleted progenitors [86]. Thus, as MABs are apparently able to cross the vessel wall, they have been used in preclinical models of systemic cell therapy for muscular dystrophy. MABs are the only cell type (in addition to myoblasts) that have been used in clinical experimentation [87].

Miraglia et al. [88] identified the expression of the CD133 antigen in hematopoietic system-derived CD34^+^ stem cells. Subsequently, CD133 expression was demonstrated in several different tissues and cells, including retinoblastoma [88], myogenic cells [89], endothelial progenitors, and fetal brain neural stem cells [90]. In particular, Torrente et al. [89] isolated and characterized a distinct stem cell population (i.e., human circulating CD133^+^ cells) that restored dystrophin expression and eventually regenerated the satellite cell pool in dystrophic scid/*mdx*. Additionally, functional tests of injected muscles revealed a substantial recovery of force after treatment. Moreover, Negroni et al. [91] also identified a highly myogenic human-muscle-derived cell type based on the expression of the cell marker CD133. They demonstrated that these cells have a much higher regenerative capacity than human myoblasts. The number of fibers expressing human proteins and the number of human cells in satellite cell anatomical positions were all increased by injection of these cells when compared with those observed after injection of human myoblasts. Additionally, CD133^+^/CD34^+^ cells exhibited an improved dispersion in the host muscle when compared with human myoblasts [91]. Finally, just like MABs, these cells can migrate through vessels [92], thus facilitating systemic delivery, which makes these cells suitable candidates for cellular therapy.

True mesenchymal stem cells (MSCs) have been isolated from adult and fetal bone marrow, but MSC-like cells are found in various tissues and organs and can differentiate into multiple cell lines [77]. Human MSC-like cells had already been isolated from healthy muscle tissue biopsies and could be were obtained using a minimally invasive biopsy procedure; skeletal muscles have thus been considered as a source of cells for therapeutic applications [93]. Gang et al. [94] isolated MSC-like cells from the umbilical cord, and they were induced to differentiate into skeletal muscle cells. These cells were able to express myogenic markers such as MyoD and myogenin [94]. Moreover, Nemeth et al. [95] showed that these cells were capable of inhibiting one of the most robust inflammatory processes, septic shock, through the modulation of macrophage activity.

In contrast to other cell-based therapies attempted for DMD patients, MSCs have the advantage of being able to fuse with and genetically complement dystrophic muscles, retain anti-inflammatory activity, and produce trophic factors that might augment the activity of endogenous repair cells [96]. De Bari et al. [97] characterized the myogenic differentiation of adult-human-synovial-membrane-derived MSCs in a *nude* mouse model of skeletal muscle regeneration, and provided proof of principle of their potential use for muscle repair in the *mdx* mouse model for DMD. In this study, cellular differentiation was shown to be sensitive to environmental cues, since MSCs injected into the bloodstream engrafted in several tissues, but acquired the muscle phenotype only within skeletal muscles. When administered into dystrophic muscles of immunosuppressed *mdx* mice, MSCs restored cytoplasmic expression of dystrophin, reduced central nucleation, and rescued the expression of mouse mechano growth factor [97].

PW1 interstitial cells were described by Mitchel et al. [82] as a population of muscle-resident stem cells located in the muscle interstitium. These cells do not express other muscle stem cell markers, such as paired box 7 molecule (Pax7), but do express the cell stress mediator PW1. These PW1^+^/Pax7^−^ interstitial cells (PICs) are myogenic in vitro and contribute to skeletal muscle regeneration in vivo, as well as generating satellite cells and more PICs. However, they have only been isolated from mouse muscles and injected intramuscularly [82].

## 4. Satellite Cells, the Main Stem Cell Target for Muscle Therapy

Muscle satellite cells are essential for skeletal muscle regeneration. They are located in muscle fibers between the basal lamina and sarcolemma as quiescent cells. After a local injury, they become activated, dissociate from the fiber, proliferate, and either undergo terminal differentiation to form new fibers or reconstitute the satellite cell pool (self-renewal cycle) [98]. Previous studies have suggested that muscle stem/precursor cells, including the myogenic satellite cell population, originate mainly from the somites, which are spheres of paraxial mesoderm that also generate skeletal muscle [99]. The most commonly used marker of SCs is Pax7 [100], and the importance of this molecule is illustrated by *Pax7* mutant mice, in which satellite cells are lacking and postnatal muscle growth and regeneration are severely compromised [101]. In *Pax7* mutants, satellite cells are progressively lost due to cellular death, with defects in the cell cycle [102]. The progression of activated satellite cells toward myogenic differentiation is controlled by a family of transcription factors (myogenic regulatory factors; MRFs), including MyoD, Myf5, myogenin, and MRF4 [103].

Sacco et al. [104] demonstrated the self-renewal capacity of satellite cells, showing that when a single muscle stem cell is transplanted into a muscle, it is capable of extensive proliferation and can be readily re-isolated, providing evidence of muscle stem cell self-renewal. SCs have also been characterized phenotypically by the expression of several surface markers, such as M-cadherin [105], CD34 [106], integrin α7β1 [107], and the chemokine receptor CXCR4 [108], among others [109]. SCs are extremely efficient at repairing muscles, and thousands of myonuclei can be generated from a small number of cells transplanted from a single fiber. Transplanted SCs can occupy their anatomical niche beneath the basal membrane and participate in future rounds of regeneration, also supporting self-renewal [110]. Since mononucleated SCs fuse to form multinucleated hybrid myotubes, sharing their nuclei in the collective gene pool syncytium, they can restore the progenitor population [73] and muscle function.

Results obtained using animal models led to clinical trials with the injection of SCs/myoblasts in phase one clinical trials. However, the growth of freshly isolated SC in vitro significantly reduces their myogenic potential, and it is thus difficult to obtain sufficient amounts of such cells. However, these trials did demonstrate that myoblast transplantation was an inefficient technique, because of the very low levels of dystrophin expression in DMD muscle fibers (approximately 1%), and there was no clinical improvement of treated patients [111,112,113]. Myoblasts are often exhausted in dystrophic conditions (such as in DMD), which hampers their isolation, modification, and amplification for autografts. Furthermore, myoblasts, whatever their origin, have limited access to target muscles when systemically injected, which is the ideal route by which to target large amounts of dystrophic muscles [68], as indicated earlier in this review. Although cellular therapies have the potential to reconstruct functional skeletal muscles and restore or improve muscle function, it is clear that many limitations must be overcome before effective therapies can be developed.

## 5. Proposals for Improvement of Stem-Cell-Based Therapies

### 5.1. Stem Cell Treatments

Muscle fibers are multinucleated cells, and the fusion of genetically corrected stem cells transferred to host muscles can generate hybrid myotubes, with exogenous genes supplying the production of functional proteins to at least partially restore normal muscle function [114]. Considering stem cells for genetic correction, expansion, and in vitro treatments for myoblast boosts (Figure 3), it is possible to generate cells capable of long-term engraftment for sustained regeneration [115], overcoming the known limitations of stem-cell-based therapies [116,117]. One basic limitation, for example, is the number of cells obtained from the muscles for in vitro expansion. To overcome this, one alternative is the conditional expression of Pax7 in human embryonic stem or induced pluripotent stem cells. In one case, the authors successfully obtained large quantities of myogenic precursors which were able to engraft efficiently upon transplantation into dystrophic muscles, producing abundant human-derived dystrophin-positive myofibers [118] (Figure 3).

In recent years, several attempts have been made to improve the culture and transplantation efficiency of myoblasts. The endogenous characteristics of these populations, including activation state and stemness, are rapidly altered in culture [104,119]. Thus, some authors have described high-yield purification processes, several different conditions for in vitro cell culture, and storage protocols for stem cell transplant [120]. In this context, muscle stem cells cultured on soft hydrogel substrates, mimicking the elasticity of muscle self-renewal in vitro, contributed to muscle regeneration when subsequently transplanted into mice [121] (Figure 3—first stage). Other published papers have attempted to mimic the SC niche using extracellular matrix (ECM) proteins such as fibronectin, which modulates cellular expansion by potentiating the Wnt7a-dependent signaling pathway [122], or the use of collagen, which readily improved the self-renewal of SC [123] (Figure 3). In another approach, the authors used nitric oxide (NO), knowing that it contributes to myogenesis and forms S-nitrosothiols (RSNO), which control signaling pathways in many different cell types. Recently, some authors showed that when inhibiting the S-nitrosoglutathione reductase (GSNOR) pathway, there was an increase in RSNO and the number of myoblasts, followed by a decrease in the myoblast fusion index. In this case, the enhanced myoblast numbers were proportional to GSNOR inhibition [124] (Figure 3). Therefore, the equilibrium between S-nitrosylation and denitrosylation with the use of inhibitors may be a useful supporting technique in culturing myoblast for transplantation. Other authors showed that myoblasts cultured in the presence of metformin, a calorie-restriction-mimicking drug, negatively regulated myogenic differentiation and slowed cell proliferation [125]. Myoblast treatment with a lower dose of metformin (Figure 3) promoted myogenic differentiation; all effects were reversible depending on the culture conditions, and there were no signs of apoptosis. Other authors showed that treating isolated skeletal muscle SCs with ursolic acid rapidly induced sirtuin (SIRT)1 expression [126]. This molecule is an essential protein involved in the regulation of cellular energy status, belonging to the sirtuin deacetylase family, and, because of its caloric restriction effects, has a vital role in longevity [127] and proliferation (Figure 3). It has also been reported that enhanced SIRT1 expression in skeletal muscle SCs stimulates their proliferation by inhibiting the expression of cell cycle inhibitors, such as the cyclin-dependent kinase inhibitors P21Waf/Cip1 and P27Kip1 [128].

Growth factors such as vascular endothelial growth factor (VEGF) reduced hypoxia-induced death of human myoblasts and improved their engraftment in mouse muscles [129]. Both insulin-like growth factor-1 (IGF-1) and basic fibroblast growth factor (bFGF) promoted the overall migration of monkey myoblasts in a serum-free environment (Figure 3), suggesting improved potential for cell dispersion after muscle injection. Moreover, treatments with these factors also stimulated components of proteolytic systems and enhanced cellular migration in vitro. As observed with human myoblasts, monkey myoblasts co-injected with these growth factors also showed increased intramuscular migration in SCID mice [130]. A short term ex vivo treatment using Wnt7a, a member of the Wingless-INT (WNT) family, on satellite stem cells markedly increased cell dispersion and engraftment, which ultimately resulted in improved muscle function of dystrophic muscles [131] (Figure 3).

To improve the efficiency of cellular transplantation, some authors have also used the strategy of co-injection of myoblasts with other cell types. Pro-inflammatory macrophages co-injected with human myoblasts improved the participation of the myoblasts in host muscle regeneration in vivo (Figure 3), increased proliferation and migration, and delayed differentiation [132]. Another paper also showed the efficiency of macrophage co-injection to improve stem cell survival, proliferation, and migration of engrafted myogenic precursor cells into *mdx* skeletal muscles [133]. Thus, the combination of multiple treatments targeting different aspects of stem cell biology in vitro (boosted stem cells) (Figure 3) may provide more appropriate cellular populations for transplant.

### 5.2. Pecipient Muscle Treatments

A promising additional strategy for improving muscle regeneration in MD is to increase the endogenous myogenic potential of the muscle environment itself for sustained regeneration after the transplant (Figure 3—second stage). Using this approach, some authors investigated the therapeutic potential of the secreted factor Wnt7a for focal treatment of dystrophic DMD muscles using *mdx* mice. They found that Wnt7a treatment efficiently induced SC expansion, myofiber hypertrophy, and increased muscle force [134]. Another approach aiming to sustain the pool of SCs in the muscles is the pharmacological manipulation of the p38 mitogen-activated protein kinase (MAPK) signaling pathway, which may restrain the decline of stem cell self-renewal in the skeletal muscles of aged mice [135]. Similarly, pharmacological inhibition of Jak2 and Stat3 activity restored the regenerative potential of SCs isolated from mice at different ages [136] (Figure 3). Several studies have also provided evidence that IL-6 may play a key role in muscle regeneration by modulating the proliferation, differentiation, and fusion of SCs to skeletal muscle fibers, indicating the importance of this cytokine to the process of muscle repair [137] (Figure 3). Moreover, regarding IL-6 production, low-level laser therapy significantly increased IL-6 mRNA 14 days after induced muscle injury [138]. Some authors showed that the treatment of SC-depleted muscles with an Activin receptor type-2B pathway inhibitor before the damage concomitantly rescued the muscles’ regenerative potential. The authors observed a near-complete inhibition of ectopic fat formation and fibrosis. They also found that the regenerated fibers originated from the residual pool of SC, suggesting that the treatment restored the regenerative potential of the few remaining cells [139] (Figure 3).

A prior requirement for the injection of boosted stem cells into pre-treated (Figure 3) damaged muscles is the correction of genetic defects. Tedesco et al. [140] reported the amelioration of the dystrophic phenotype in *mdx* mice transplanted with murine muscle progenitors containing a human artificial chromosome with the entire dystrophin locus. Recently, other authors showed that a reversibly immortalized cell population was able to extend cellular proliferation; these SC-derived myoblasts and perivascular-cell-derived MABs were also transferred with the dystrophin locus in a human artificial chromosome [141]. These genetically corrected cells did not undergo tumorigenic transformation, retained their migration ability, and remained myogenic in vitro. Additionally, Crist et al. [142] showed that muscle stem cells modified by microRNA-27 regulated the expression of Pax3, delayed SC differentiation, and downregulated SC entry into the myogenic differentiation program. Later, other authors showed that the MyoD family inhibitor (MDFI) is the miRNA-27 target gene [143]. Therefore, these results indicate that miRNAs and genes work collaboratively in regulating SCs, which may be useful in future transplant trials. These approaches to genetically correct pluripotent stem cells can also be made in autologous cells for transplantation into MD patients [144]. Additionally, frameshifting, exon knock-in, or exon skipping can be used for patient-derived human induced pluripotent stem cells using CRISPR/Cas9 technology [144,145]. It has recently been published that a single, clustered, regularly interspaced short palindromic repeat (CRISPR)/Cas9 cleavage in either its 50 or 30 unique flanks promotes the deletion of large segments of the repeat sequence [146] (Figure 3). This research was done using myoblasts from unaffected individuals, MMD1 patients, and an MMD1 mouse model, and led to a very precise excision of the repeat tract, with no apparent effects on the expression of the MMD1 locus genes. Importantly, the authors stated that the myogenic capacity, nucleocytoplasmic distribution, and abnormal RNP-binding behavior of transcripts from the edited *DMPK* gene were normalized. Therefore, cell line banks maintaining and providing isogenic progenitor cells, autologous or not, bearing corrected genes may be a solution for many MD patients.

Although cellular-therapy-based clinical trials for DMD have not advanced in terms of benefit to the patients, the recent clinical success of autologous myoblast therapy for OPMD pharyngeal muscles shows that cellular therapy may still hold new therapeutic strategies. One study that reached the clinical phase I/IIa included 12 OPMD patients and assessed the practicability of myoblast transplantation in pharyngeal muscles. The results of this trial showed short- and long-term (2 years) safety and tolerability in all patients with no adverse effects, and there was an improvement in the quality of life score for all treated patients [147].

Many treatments for in vitro stem cell boosting and in vivo target muscle pre-treatments are yet to be discovered or tested, in addition to the ones illustrated in this review. A large family of biologically active peptides is of particular interest, namely the matricryptins. These small molecules were previously named cryptic fragments and have been studied in the field of cancer research for many years. These fragments induce arrest of cellular proliferation or mitosis, migration in culture, alteration in gene clusters transcription, cell death, cell adhesion, cytoskeleton rearrangement, activation of focal adhesion kinases (FAKs), and many other processes (reviewed in Reference [148]). Different matricryptins can be used in combination with other treatments for stem cell and target muscle boosts, as long as many of these extracellular matrix components (ECM) fragments have biological features that complement the needs and limitations of cellular therapies.

## 6. Matricryptins as Possible New Players in Stem Cell Therapy

Matricryptins are fragments released from ECM proteins [149]; the name was proposed by Davis et al. [150], who described these biologically active peptides as “enzymatic fragments of ECM containing exposed matricryptic sites”. Moreover, the mechanisms that regulate the exposure of matricryptic sites “include ECM components multimerization, heterotypic binding, adsorption to other molecules, cell-mediated mechanical forces, exposure to reactive oxygen species, and ECM denaturation”, for example. Matricryptins expose matricryptic sites [149] generally not presented in the original full-length molecules [150,151]. Concerning their function, they regulate numerous biological processes in physiological and pathological situations, such as autophagy, angiogenesis, adipogenesis, fibrosis, tumor growth, metastasis, and wound healing. Matricryptins have various molecular functions and modulate gene expression, cell signaling, or act as enzymes, proenzyme activators, or enzyme inhibitors [152]. In this way, matricryptins can be considered potential drugs. Endostatin, for example, has been approved in China for the treatment of small-cell lung cancer in combination with chemotherapy, and has been tested in several clinical trials. Endostatin is a naturally occurring 20 kDa C-terminal fragment derived from collagen type XVIII which acts as a potent anti-angiogenesis factor [153].

One of the main sources of matricryptins is laminin-111—a major component of the basement membrane matrix with diverse biological functions, it is a trimeric glycoprotein composed of α, β, and γ chains. Laminin-111 is the most well-studied component of some 15 laminin isoforms because it can be isolated in quantity from mouse Engelbreth–Holm–Swarm (EHS) tumors and is commercially available. Laminin-111 interacts with cells and has multiple biological activities, including neurite outgrowth, tumor metastasis, cell attachment and spreading, and angiogenesis [154]. Laminin-derived peptides are involved in various different biological activities, and several active sequences in laminin-111 have been identified using systematic screening of synthetic peptides [155]. In these studies, several peptides (matricryptins) with cellular adhesive properties were identified, among them the AG73 (RKRLQVQLSIRT, mouse laminin α1 chain 2719–2730), which showed the most potent cell attachment activity among the G domain peptides and promoted neurite outgrowth [155,156]. The Ag73 peptide has also been described as having angiogenic activity in several systems, and has already been indicated in tissue engineering and regeneration due to its ability to promote cell adhesion [157,158]. The peptides A13 and C16, which are present in the homologous NH(2)-terminal domains of the α1 and γ1 chains of laminin, respectively, also showed significant in vivo angiogenic activity and wound healing [159,160]. In addition to affecting angiogenesis, the C16 peptide also both promoted B16F10 melanoma cell migration in vitro and enhanced pulmonary metastases in vivo. Since C16 induced the production of membrane metalloproteinase 9 (MMP-9) by these cells, it is clear that this site on the γ1 chain is essential in tumor cell metastasis as well as in angiogenesis. Another well-studied bioactive peptide derived from laminin is the sequence SIKVAV; this laminin-α1 chain peptide was initially described as promoting cell adhesion, migration, and neurite outgrowth, but it was soon found to be a potent stimulator of tumor growth, metastasis, protease activation/secretion, and angiogenesis [161,162]. Similarly to the SIKVAV peptide, AG73 increased subcutaneous tumor growth and lung colonization of B16F10 melanoma cells. This peptide also induced B16F10 liver metastasis [163]. The identification and study of these cryptic peptides and many others derived from laminin will be important in defining therapeutic strategies for cancer and many other diseases.

Although studies using matricryptins have not yet been published in the field of muscle stem cells, the biological activity of these peptides in tumor cells make them a promising group of molecules for muscle stem cell boosts. Laminin cryptic fragments have been used in experiments conducted in our laboratory with excellent results, increasing the proliferation, migration, and survival rates of myoblasts used in experimental transplantation. It is interesting to note that some individual matricryptins tested performed distinct biological functions when comparing cancer cells and myoblasts (data not shown). These results open new avenues for cultured stem cell boosts before transplantation and receipient muscle pre-treatment with matricryptins, combined with other described boosts.

We tested some laminin-derived matricryptins in the proliferation, migration, and terminal differentiation of primary myoblasts and the C2C12 myoblast cell lineage. We also treated some target muscles after induced muscle damage, such as in the tibialis anterior, and observed promising results in in vivo regeneration and inflammatory control. The distinct biological capacity of AG73 and SIKVAV peptides over myoblast chemotaxis/migration in a transwell assay is illustrated in Figure 4, where we show that laminin and laminin-derived fragments have different impacts on C2C12 migration.

## 7. Concluding Remarks

In this review, we aimed to highlight the enormous beneficial potential of stem-cell-based therapies, discussing different cell types and biological characteristics. The challenges to be overcome before practical clinical use of stem cells can recover patients’ life quality and expectancy are equally enormous. However, the recent advances made and knowledge gathered regarding stem cell biology and function are significant, and may be converted into effective clinical treatments. Considering the multiple types of MD, different stem cells may be isolated and expanded, and different in vitro treatments and culture conditions can be combined for stem cell boost and reimplantation or transplant. It is essential to indicate that direct cellular manipulations can correct the underlying genetic defects that lead to MDs (Figure 3), in contrast to much more sophisticated approaches for in vivo genetic corrections.

For successful use of muscle stem cells, combined in vitro boosts should improve the cellular capacity for proliferation, survival, dispersion throughout the muscles, fusion, differentiation in functional muscle fibers, and ability to repopulate muscle stem cell niches to continually provide corrected myoblasts (Figure 3). MD patients affected by restricted muscle groups are more likely to benefit from boosted stem cell transplants, and matricryptins are possible players in this scenario. Matricryptins associated with other in vitro treatments may induce critical biological responses in muscle stem cells, as previously demonstrated in cancer cells. Moreover, in vivo treatment of target muscle groups before stem cell transplant may also favor engraftment (Figure 3), preparing the muscle microenvironment for new cycles of sustained regeneration. In summary, our current knowledge presents new and exciting perspectives for future treatments for MD patients.

## Figures and Tables

**Figure 1 ijms-20-05433-f001:**
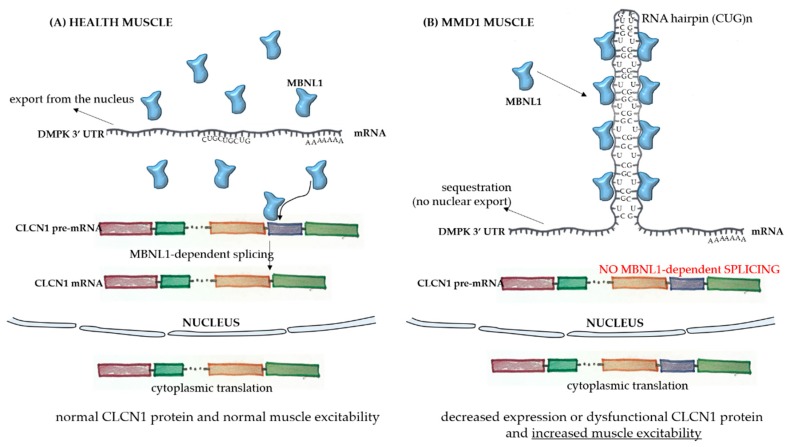
Pathophysiology of myotonic muscular dystrophy type 1. In healthy skeletal muscle fibers (**A**), the mRNA sequence of the dystrophia myotonica protein kinase (*DMPK*) gene has a limited number of the repeat sequence CUG, and there is normal export of the RNA to the cytoplasm. The MBNL1 factor is available to play its role in the processing of other transcripts, such as the pre mRNA of CLCN1, in the nucleus. After correct splicing, mature mRNAs are exported to the cytoplasm for protein translation and normal function in subcellular compartments. In the case of muscle fibers from myotonic muscular dystrophy type 1 (MMD1) patients (**B**), there is an aberrant expansion of the CUG repeat in the *DMPK* mRNA, with the formation of a hairpin due to the pairing of the bases composing the repeats. In this case, there is sequestration of the MBNL1 factor, which binds to the hairpin with its sequestration in the nucleus. The complex then precipitates into microscopically visible nuclear foci. Because of the MBNL1 retention, the nuclear splicing of pre mRNAs is compromised, leading to decreased or dysfunctional protein production.

**Figure 2 ijms-20-05433-f002:**
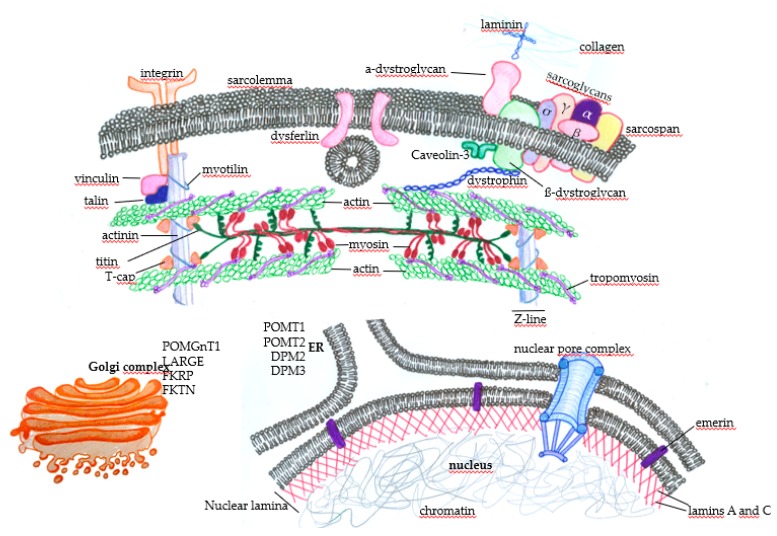
Subcellular distribution of some proteins that, when genetically altered, lead to muscular dystrophies. In the sarcolemma, there are proteins, such as the components of the dystrophin-associated glycoprotein complex (DGC) (α-dystroglycan, σ, γ, β, and α sarcoglycans, sarcospan, and β-dystroglycan), that physically mediate the interaction of the extracellular ambience with the cytoplasm through the binding of laminin with extracellular matrix components such as collagen. The protein dystrophin mediates the integration of the complex with actin filaments, composing the contractile structures of the muscle cytoskeleton. The concerted action of these proteins protects the muscle fibers against cellular damage during the cycles of contraction. Membrane integrins also interact with extracellular matrix components and provide normal muscle function and contraction through the integration with intracellular components. Under physiological circumstances, when there is the disruption of the sarcolemma, the action of dysferlin mediates membrane repair to avoid cell death. Many proteins play essential roles in maintaining normal muscle function and contraction force, such as vinculin, talin, and myotilin, alongside titin, T-cap (composing the Z line), tropomyosin, and the main contractile components, myosin and actin. Nuclear proteins such as emerin, which anchors the nuclear lamina components lamin A and C, can also induce muscular dystrophies when genetically mutated. Finally, alterations in the glycosylation patterns of proteins such as (protein o-mannosyltransferase) POMT1, POMT2, (dolichol-phosphate-mannose) DPM2, and DPM3 in the endoplasmic reticulum (ER) and (protein O-mannose b-1, 2-N-acetylglucosaminyltransferase) POMGnT1, (like-glycosyltransferase) LARGE, (Fukutin-related protein) FKRP, and (Fukutin) FKTN in the Golgi complex deeply affect muscle function and lead to muscular dystrophy.

**Figure 3 ijms-20-05433-f003:**
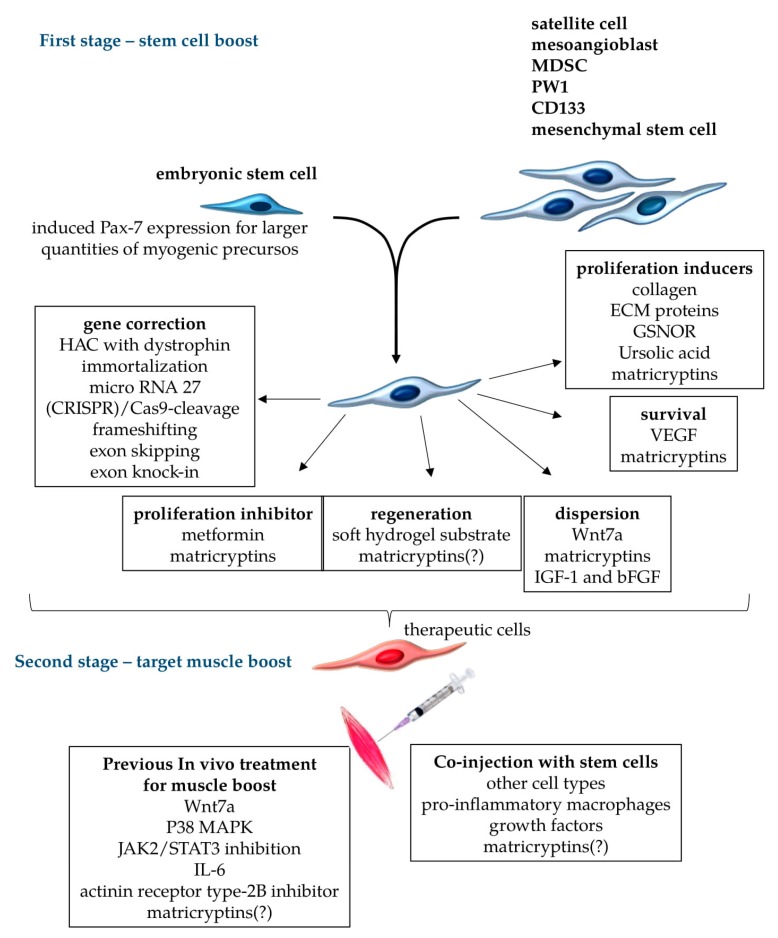
Treatments for stem cells and target muscles to improve stem-cell-based cellular therapy proposals. Different cell types can be used for therapeutic transplants in muscular dystrophies, including embryonic stem cells with the induced expression of (paired box protein) Pax7 for myogenic commitment, and muscle cells such as satellite cells, mesoangioblasts, (muscle-derived stem cell) MDSC, PW1, CD133, and mesenchymal stem cells. For transplant therapy, appropriate stem cells should be submitted to gene correction procedures to restore the normal production of functional, unmutated proteins. Further cellular boost could aim to reversibly inhibit proliferation (when terminal differentiation is necessary), induce terminal muscle differentiation and dispersion of stem cells through the treated muscles, increase the survival rate of transferred cells, and induce proliferation to expand the population of cells to be engrafted. These cellular in vitro treatments comprise the first stage of the boosts, which can be done in different combinations to improve engraftment, focusing on particular necessities of each muscular dystrophy. The second stage of treatment should target the muscles that receive boosted cells. In this case, the muscles can be pre-treated with chemical metabolic regulators to facilitate cellular engraftment, or receive the co-injection of other cell types, growth factors, and other components to improve the transplant. Matricryptins are a family of pleiotropic biologically active peptides derived from extracellular matrix components that serve many of these functions, as observed in tumor cells. Matricryptins are described as modulators of cellular migration, proliferation, adhesion, and cell survival. It is yet to be shown whether matricryptins can induce the regeneration of stem cells into muscle fibers, and the nature of their activities in the environments of different muscles (indicated by “?” in the scheme).

**Figure 4 ijms-20-05433-f004:**
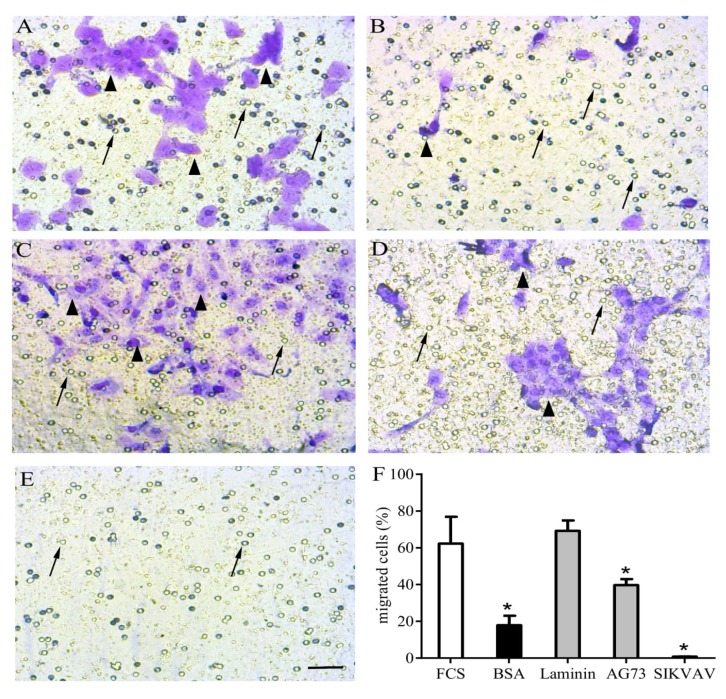
Matricryptins and in vitro myoblast chemotaxis/migration. Laminin-111 and laminin-derived peptides were used to evaluate C2C12 myoblast chemotaxis/migration assayed in a transwell system. The cells were plated over a membrane (8 µm pores) and fetal calf serum (FCS) 20% was used as a positive migration control in the lower chamber (**A**). All other stimuli were applied in the absence of FCS, with highly purified bovine serum albumin (BSA) as a negative control (**B**), full length laminin-111 (**C**), and the laminin-111-derived peptides AG73 (RKRLQVQLSIRT) (**D**) and SIKVAV (**E**), all at concentrations of 100 µg/well. After a 12 h incubation, the unresponsive cells were gently removed from the membrane’s upper surface, and the cells that migrated were fixed and stained using crystal violet diluted in ethanol. The graphic shows the relative number of cells that migrated towards the lower chamber according to the indicated stimulus (**F**). The myoblasts are indicated by arrowheads, and arrows indicate membrane pores. * means p<0.05. Bar: 40 µm.

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
