# Peer review of "A Promising Future for Stem-Cell-Based Therapies in Muscular Dystrophies—In Vitro and In Vivo Treatments to Boost Cellular Engraftment"

_ijms, 2019, doi:10.3390/ijms20215433_

Round 1
Reviewer 1 Report
In the manuscript ‘A Promising Future for Stem Cell-Based Therapies in Muscular Dystrophies, matricryptins as new players’ the authors aim to review currently available genetic, biochemical and cell culture treatments that can boost muscle stem cell engraftment, with particular reference to the use of matricryptins. Matricryptins and matrikines are small biologically active fragments released when extracellular matrix proteins are digested by enzymes. They are known to have roles in tumorigenesis and tissue remodeling.
The authors of this manuscript have written an extensive review of the clinical symptoms and genetic causes of a range of muscular dystrophies and provide a nice overview of the types and limitations of cell therapies to reduce clinical symptoms. However, I have some major reservations about the content of this review which I outline below.
Major issues:
1. Following a quick PubMed literature search, I was surprised to find that there are currently no papers looking at the effect of matricyptins and matrikines on skeletal muscle or myoblast function. Therefore, there is no peer-reviewed evidence base that matricryptins/matrikines have a role in muscle regeneration, or could contribute to improving myogenic cell therapies – everything in this review is therefore speculative! The authors do show in figure 4 that they have some preliminary unpublished data on the effect of one matricryptin in C2C12 myoblasts (an immortalized myoblast line). Here they look at the effect of laminin-111 fragments on C2C12 myoblast migration. There are several issues with this. Firstly, it is a strange choice of experiment as myoblasts do not normally migrate through small spaces (this experiment is most often used to model trans-endothelial migration) so the reasoning as to why a transwell assay was performed is unclear. Secondly, this preliminary and unpublished experiment is not enough to suggest that matricryptins effect muscle progenitor function and definitely is not enough to suggest any of the future therapeutic avenues for matricryptins role in stem cell therapies for muscular dystrophies. This review is premature and hypothetical, so therefore unfortunately, I cannot support it.
2. The manuscript includes overcomplicated phrasing, long sentences with overuse of commas and semi-colons, spelling mistakes and the interchangeable use of abbreviations. These make the manuscript difficult to read. I would recommend the use of an English-language editor.
3. The section about different forms cell transplantation is not clear and is not referenced with examples to back up the claims. For example, the authors reference two papers that show that injecting macrophages with myoblasts improves myoblast engraftment (line 555 onwards). However, the authors extend this (within both the text and figure 3) to include ‘other cells’, ‘growth factors’ and ‘matricryptins’ without providing the evidence base. Other examples can be found throughout the text.
4. There is a lack of clarity surrounding transplantation routes; the authors claim that intra-arterial transplantation of myoblasts is a better route for systemic injection than IV. However, myoblasts cannot be used for systemic transplantation for either route! Systemic transplantation of myogenic cells is only possible using cells that normally circulate i.e CD133+ cells or mabs not satellite cell-derived myoblasts. For this reason, myoblasts are no longer the main target for cell therapy delivery, unless it’s for specific muscles that can be delivered via the intra-muscular route.
Minor:
- lines 35-7 ‘9 significant forms of MD’. Many would disagree that only 9 forms of MD are significant – do you mean the most common? please change wording.
- Table 1 (supp): FSHD affected protein should be ‘a toxic-gain-of-function DUX4 protein’, not subteleomic chromatin rearrangement
- Line 44: use of ‘and others’ is unclear in this context.
- Whilst the figures nicely show disease-relevant molecules, there are several spelling mistakes. For example, in figure 1 ‘heath’ muscle should be ‘healthy’ and in figure 2 ‘distrophin’ is spelt ‘dystrophin’. Please check over.
- The disease name is Emery-Dreiffus, not Emery and Dreiffus
Author Response
"Please see the attachment."

Reviewer 2 Report
The authors describe the majority of types of MD and then review the treatments that utilize a variety of stem cells. The review provides a good overview of the material, but can be improved.
The review would benefit from an introductory paragraph describing the approach and a well defined aim of the review.
The authors spend a large amount of space defining the different types of MD. While the information is interesting, it does not appear to be necessary for the review. There is no specific instances in the second portion of stem cell therapy that requires the reader to know the specifics of each disorder. If the authors feel compelled to leave this massive amount of information, they need to make it relevant in the treatment sections. For example, do matricryptins pose a better drug target in specific types of MD?
The use of subheadings, especiially in section 5. Proposal for stem-based cell therapies improvments would help improve the readers experience.
Since the title of the article is about Matricrptins, that authors need to highlight the pros/cons of these fragments
Author Response
Dear editor of the International Journal of Molecular Sciences
We are deeply thankful for the reviewers’ comments, suggestions, and critics. We believe that the manuscript is substantially improved after the revision and hope that the final version can be accepted by this prestigious Journal. Please feel free to contact us if the issues raised by the reviewers were not adequately addressed and answered in this version. We will be more than happy to make further adjustments.
Response to Reviewer 2 Comments
Point 1: The authors describe the majority of types of MD and then review the treatments that utilize a variety of stem cells. The review provides a good overview of the material, but can be improved.
The review would benefit from an introductory paragraph describing the approach and a well defined aim of the review.
Response 1: We agreed with this comment and included at the end of the Introduction the following paragraph:
“ In this review, we will summarize the most common forms of MD and their current treatments. However, most patients have only palliative therapeutic strategies that aim to alleviate the symptoms, with no effective treatments available. Although for all MD patients, stem cell-based therapies pose as a hope for better life quality and expectancy, this alternative has many limitations. Stem cells usually have limited capacity to engraft in the muscles due to reduced cellular viability, dispersion, proliferation, and differentiation to myotubes, for example. Many of these limitations have individually been addressed in the literature, with biochemical, genetic and in vitro culture approaches that can improve cellular engraftment. Here we summarize these stimuli and propose combined treatments for the stem cells and the recipient muscles aiming to prolong grafted cells survival and sustained health myotubes formation. Although not yet tested in muscle stem cells, matricryptins are nontoxic bioactive peptides that induce tumor cells survival, migration, proliferation, and differentiation wanted biological responses for stem cells. We discuss these molecules as possible adjuvants that can be used in combination with other stimuli for stem cells boost in muscle therapy.”
Point 2: The authors spend a large amount of space defining the different types of MD. While the information is interesting, it does not appear to be necessary for the review. There is no specific instances in the second portion of stem cell therapy that requires the reader to know the specifics of each disorder. If the authors feel compelled to leave this massive amount of information, they need to make it relevant in the treatment sections. For example, do matricryptins pose a better drug target in specific types of MD?
Response 2: We agree that the section regarding the different types of MD is extended; however, we intended to provide a summary of the main aspects of each MD for readers unfamiliar with this subject. It was also the intention when we prepared the Supplementary Table 1. We believe that it is important to make clear that MD pathogenesis is rather vast, with numerous genetic defects and different consequent cellular alterations and symptoms. This broad clinical spectrum justifies the proposal that different combinations of stem cells and recipient muscle treatments can be suitable for various diseases.
As we answered to reviewer #1, matricryptins are proposed as a group of peptides that can potentially lead to good results. This indication is based on the biological responses induced by matricryptins over tumor cells, leading to proliferation or mitosis arrest, migration, survival, cell differentiation, and others. Collectively, these biological responses are expected from stem cells used for cellular therapies, and we propose that these peptides can be tested in MD. We have promising results regarding myoblasts and matricryptins in our laboratory, which are composing another manuscript. To illustrate that these molecules act over myoblasts, we inserted the Fig. 4 in the manuscript.
To make this point clearer, we included the following sentence by the end of the Introduction: “The broad spectrum of pathogenic genetic defects and clinical symptoms suggests that different combinations of boosts may be required for different stem cell-based therapies.”
Point 3: The use of subheadings, especially in section 5. Proposal for stem-based cell therapies improvments would help improve the readers experience.
Response 3: We separated it into two items.
Point 4: Since the title of the article is about Matricrptins, that authors need to highlight the pros/cons of these fragments
Response 4: This is true, and we changed the title and Abstract accordingly. The review summarizes different in vitro and in vivo (pre)treatments and proposes that the combination of these treatments may improve the engraftment of stem cells in MD patients. Considering the desirable biological effects of these treatments, as proliferation, migration, and differentiation, we propose matricryptins as a new group of molecules that may be tested. We believe that this idea is more evident in the second version of the manuscript.
Reviewer 3 Report
This is a comprehensive review, very well written and pleasant to read. A detailed classification of muscular dystrophies is followed by a dissertation about stem cell-based therapies and their potential improvements. The reference list is appropriate.
Minor points
Line 203. Please replace “Gystrophy” with “Dystrophy”. Line 266. Please replace “muscular” with “Muscular”. Line 404. Please replace “MSDCs” with “MDSCs”. Line 468. Please use the singular in the sentence, “The muscle satellite cells are essential for skeletal muscle regeneration.” to concord with the subsequent sentences, which use the singular. Lines 504,505. The authors state, “Muscle fibers are multinucleated cells, and the fusion of genetically corrected stem cells transferred to host muscles generates hybrid myocytes,” however the term “myocytes” indicates mononuclear cells in the terminal phase of the muscle differentiation process, just before fusion into myotubes or immature myofibers. Please correct that sentence accordingly. Line 528. Please delete “;”. Line 593. Please replace “maybe” with “may be”. Line 629. Please replace “[150], [151]” with “[150,151]”. Figure 4. C2C12 myoblasts should be treated with the same molarity of the different factors rather than the same amount (micrograms/well) due to the very different molecular weight of the factors tested. A graphical representation should be added reporting the average numbers of cells passed through the porous membrane in the conditions tested.Author Response
"Please see the attachment."

Round 2
Reviewer 1 Report
I have re-read the new manuscript for the Beghini et al review following initial peer review. In summary, the article is now much clearer, accurate and interesting to read. I appreciate their efforts to address the issues brought up in the initial review; specifically, I thank the authors for changing the wording (and meaning) which implied that matricryptins have a known role in regeneration. Also, the re-wording of the transplantation section is much clearer (points 3 + 4). There is only I small issue that needs to be addressed regarding spelling errors, as highlighted below:
Figure 1: spelling errors remain for ‘(A) health muscle’ (should be ‘healthy muscle’), and ‘dependente’ (should be ‘dependant’)
Figure 3 spelling errors: ‘ ‘mesenchimal’ (should be ‘mesenchymal’), ‘stem cells boost’ (should be ‘stem cell boost’)